# Panoramic Mapping with Information Technologies for Supporting Engineering Education: A Preliminary Exploration

**Jhe-Syuan Lai \*, Yu-Chi Peng, Min-Jhen Chang and Jun-Yi Huang**

Department of Civil Engineering, Feng Chia University, Taichung 40724, Taiwan;
M0702975@mail.fcu.edu.tw (Y.-C.P.); d0676303@mail.fcu.edu.tw (M.-J.C.); M0806485@mail.fcu.edu.tw (J.-Y.H.)
\* Correspondence: jslai@fcu.edu.tw; Tel.: +886-4-2451725 (ext. 3118)

**Abstract:** The present researchers took multistation-based panoramic images and imported the processed images into a virtual tour platform to create webpages and a virtual reality environment. The integrated multimedia platform aims to assist students in a surveying practice course. A questionnaire survey was conducted to evaluate the platform's usefulness to students, and its design was modified according to respondents' feedback. Panoramic photos were taken using a full-frame digital single-lens reflex camera with an ultra-wide-angle zoom lens mounted on a panoramic instrument. The camera took photos at various angles, generating a visual field with horizontal and vertical viewing angles close to 360°. Multiple overlapping images were stitched to form a complete panoramic image for each capturing station. Image stitching entails extracting feature points to verify the correspondence between the same feature point in different images (i.e., tie points). By calculating the root mean square error of a stitched image, we determined the stitching quality and modified the tie point location when necessary. The root mean square errors of nearly all panoramas were lower than 5 pixels, meeting the recommended stitching standard. Additionally, 92% of the respondents ($n = 62$) considered the platform helpful for their surveying practice course. We also discussed and provided suggestions for the improvement of panoramic image quality, camera parameter settings, and panoramic image processing.

**Keywords:** engineering education; image stitching; information technology; multimedia; panorama

## 1. Introduction

Panoramic images are being made available on an increasing number of online media platforms, such as Google Maps. Virtual reality (VR) technology is also becoming more common in modern life (e.g., video games and street view maps), providing immersive and interactive experiences for users. Various industries have incorporated this technology into their businesses; for example, companies in the leisure industry, such as the Garinko Ice-Breaker Cruise 360 Experience in Hokkaido, include VR images on their official websites to attract tourists [1]. Similarly, several real estate companies are showcasing furnished interior spaces to potential buyers by using panoramic images; this helps customers visualize the actual setting of the houses they are interested in [2]. These multimedia approaches integrating image, video, audio, and animations can further obtain better presentation and communication methods [3–7], such as visualization, map, graphical user interfaces, interactive recommendation system, etc.

Surveying practice is a fundamental and essential subject for civil engineering students. However, university students, with little civil engineering experience, mostly do not know how to accomplish a surveying task because of (a) unfamiliarity with surveying points, (b) inability to

connect knowledge acquired in class with actual practices, (c) inability related to or unfamiliarity with establishing a record table, and (d) inability to operate or unfamiliarity with instrument operations. Lu [8] studied computer-assisted instruction in engineering surveying practice. According to the questionnaire survey results, 91% of the students either strongly agreed or agreed that virtual equipment helped increase their learning motivation. Additionally, 66% of the respondents correctly answered a question concerning azimuth, a foundational concept in civil engineering. Lu [8] indicated that the introduction of digital instruction is more likely to spark learning interest and motivation than conventional training would.

By integrating panoramic images into a virtual tour platform, this study adopted VR technology to create a webpage that facilitates the instruction of surveying practice. The present researchers selected panoramic images because of the low cost of image construction and ability to create realistic and immersive visual effects. The designed assistance platform included (a) surveying tips, (b) various surveying routes, (c) corresponding measurement principles, and (d) instructional videos. Therefore, students had access to the supplementary materials on this platform before or during a lesson, thereby increasing learning efficiency and helping students acquire independent learning skills. This study explored student acceptance of technology-aided instruction and the practicality of such an instruction method by using a questionnaire survey. Subsequently, the original web design was modified per the students' feedback. In this paper, the authors also discussed and proposed suggestions for the improvement of panoramic image quality, camera parameter settings, and panoramic image processing.

## 2. Related Works

### 2.1. Panorama

The word "panorama" originates from the Greek pan ("all") and horama ("view"). In 1857, M. Garrela patented a camera in England that could rotate around its own axis and take a horizontal 360° photo; it was the first camera for panoramic photos that employed mainspring control. According to the field of capture, panorama can be divided into three patterns, as presented in Table 1. In this study, the shooting targets were all objects above the ground, and the angle of coverage was mainly landscape; thus, each shot did not fully cover the landscape in the vertical direction. According to the range and angle settings, the photos taken in this study are considered 360° panoramas.

**Table 1.** Classification of Panorama [9].

| Field of Capturing | Panorama | 360° Panorama | Spherical Panorama |
|---|---|---|---|
| Horizontal | <180° | 360° | 360° |
| Vertical | <180° | <180° | 180° |

Image stitching is a crucial step in panorama generation. Zheng et al. [10] described and explored shooting strategies and image stitching methods in detail. Regarding research on the stitching process, Chen and Tseng [11], by identifying the corresponding feature point in different images (i.e., tie point), determined the tie point quality of panoramic images. By using panoramic photography and photogrammetry, Teo and Chang [12] and Laliberte et al. [13] generated three-dimensional (3D) image-based point clouds and orthophotos. Studies have also applied image stitching in numerous areas, including landscape identification, indoor positioning and navigation, 3D city models, virtual tours, rock art digital enhancement, and campus virtual tours [14–19].

### 2.2. Virtual Reality (VR)

VR is a type of computer-based simulation. Dennis and Kansky [20] stated that VR can generate simulated scenes that enable users to experience, learn, and freely observe objects in a 3D space in real time. When users move, the computer immediately performs complex calculations and

returns precise 3D images to create a sense of presence. VR integrates the latest techniques in computer graphics, artificial intelligence, sensing technology, display technology, and Internet parallel computing. Burdea [21] suggested defining VR according to its functions and proposed the concept of three Is (i.e., interaction, immersion, and imagination), and suggested that VR must have said three characteristics. Furthermore, Gupta et al. [22] connected VR with soundscape data.

VR devices can be either computer-based (high resolution) or smartphone-based (portable). For example, the HTC VIVE-Pro is computer-based, whereas Google Cardboard is smartphone-based. We selected the portable VR device to conduct experiments because it is inexpensive and does not require a computer connection. To access the designed online assistance platform for teaching, the participating students only had to click on the webpage link or scan the quick response (QR) code using their smartphones.

## 2.3. Education with Information Technologies

As a result of technological advancement, e-learning has become prevalent. For example, information technologies, such as smartphones, multimedia, augmented reality, and internet of things, have been adopted to increase or explore learning outcomes [23–26]. Lee [27] maintained that through learners' active participation, computer-based simulation can effectively assist learners to understand abstract concepts, which in turn increases learning motivation and improves learning outcomes. VR can also help create a learning environment without time constraints; for example, Brenton et al. [28] incorporated VR into anatomy teaching to mitigate the major impediments to anatomy teaching, such as time constraints and limited availability of cadavers, by using 3D modeling as well as computer-assisted learning. The Archeoguide (Augmented Reality-based Cultural Heritage On-site Guide) proposed by Vlahakis et al. [29] demonstrated that VR is not bounded by spatial constraints. This on-site guide was used to provide a customized cultural heritage tour experience for tourists. ART EMPEROR [30] discovered that numerous prominent museums worldwide have established databases of their collections using high-resolution photography or 3D scanning and modeling. These databases enable users to explore art through the Internet.

Chao [31] surveyed and conducted in-depth interviews with VR users after they participated in VR-related scientific experiments; the results indicated that such experiments provide participants with the illusion that they are in a physical environment. Without spatial constraints, the VR-simulated environment offered the participants experiences that they could not have in real life. Therefore, VR helped the participants obtain information and knowledge of various fields in a practical manner. These experiences, compared with those obtained through videos and print books, left a stronger impression on students, prompting them to actively seek answers. Similarly, Chang [32] indicated that students receiving 3D panorama-based instruction significantly outperformed their counterparts who received conventional instruction. Liao [33] examined English learning outcomes and motivation among vocational high school students by using panorama and VR technology; the results revealed that the use of said technologies effectively improved learning outcomes, motivation, and satisfaction.

## 2.4. Summary

According to the aforementioned literature, panoramic photography and VR technology have advanced rapidly, have a wide range of applications, provide realistic 3D experiences, and enhance teaching effectiveness. However, few studies have applied said technologies to engineering education. The present study created panorama-based VR environments on a virtual tour platform and achieved a cost-effective display of on-site scenes. The research team hopes to help civil engineering students rapidly become familiar with the surveying practice elements in question and complement theories with practical knowledge and skills.

## 3. Materials and Methods

Figure 1 presents the study procedures. First, capturing stations were set up and images were collected (Section 3.1). Subsequently, image stitching was performed (Section 3.2) by combining multiple photos from a single station into a panoramic image. The combined panoramic images were then transformed into web format by using a virtual tour platform (Section 3.3); additional functions could be added to the web format. Next, a VR navigation environment was constructed, finalizing the development of a teaching assistance platform. The platform was assessed using a questionnaire survey (Section 3.4); the survey results and feedback from users were referenced to improve the platform design.

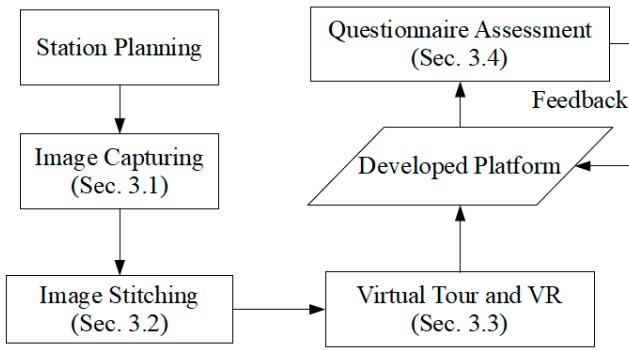

**Figure 1.** The conceptual procedure used in this study.

### 3.1. Image Capturing

The researchers mounted a full-frame digital single-lens reflex camera (CANON EOS 6D Mark II, Canon, Tokyo, Japan) equipped with an ultra-wide-angle zoom lens (Canon EF 16–35 mm F/4L IS USM, Canon, Tokyo, Japan) on a panoramic instrument (GigaPan EPIC Pro V, GigaPan, Portland, OR, USA; Figure 2) and tripod. To use the GigaPan, the horizontal and vertical coverage must be set, after which the machine automatically rotates and accurately divides the scene into several grid images; these images overlap, which facilitates the image stitching process. In addition, the camera took each photo by using bracketing and captured shots at three brightness levels (normal, darker, and brighter). These shots served as material for stitching and synthesis. Please refer to https://www.youtube.com/watch?v=JTkFZwhRuxQ for the actual shooting process employing the GigaPan.

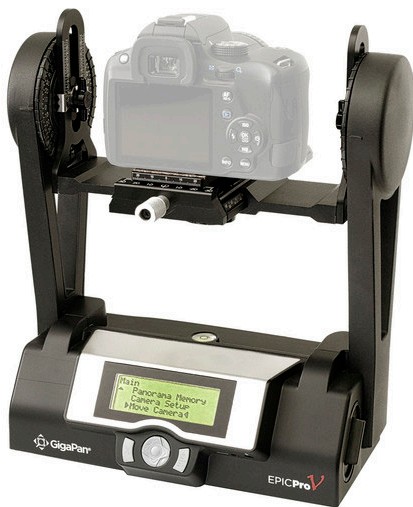

**Figure 2.** Panoramic instrument of GigaPan EPIC Pro V.

### 3.2. Image Stitching

After we captured the images, Kolor Autopano was adopted in this study. The original images underwent feature point extraction, homography, warping, and blending to form the panoramic image of a station. The general process is detailed in Figure 3.

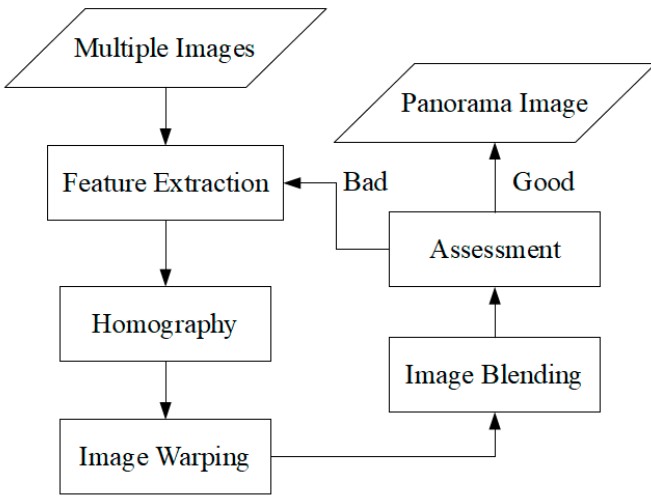

**Figure 3.** Procedure for image stitching.

### 3.2.1. Homography

When two images partially overlap, they share several corresponding feature points. These points can be connected using computations; this process is known as homography, and the corresponding feature points are called tie points.

Kolor Autopano extracts, matches, and transforms feature points into tie points by using the scale-invariant feature transform (SIFT) algorithm [10]. The features extracted using this technique are invariant to image rotation and scaling as well as changes in grayscale values. There are four major steps, including (a) scale-space extrema detection, (b) key-point localization, (c) orientation assignment, and (d) key-point descriptor. The results of detecting points of interest are termed as key-point candidates in the step of scale-space extrema detection. The Gaussian filter was used to convolve at different scales, and difference of Gaussian-blurred images were obtained. Key-points of the maximum and minimum Difference of Gaussians (DoG) are further extracted at multiple scales. A DoG image (D) is given by Equation (1), where L represents the convolution of the original image (x,y) with the Gaussian blur at scales of kσ; k and σ indicate a scale factor and standard deviation of the Gaussian blur, respectively. The second step is to localize the key-points. The scale-space extrema detection might produce too many unstable key-point candidates. This step is to fit the nearby data for accurate location in consideration of scale and ratio of principal curvatures. For assigning the orientation of key-points, the local image gradient directions in achieving invariance to rotation were determined by Equations (2) and (3), where θ and m represent orientation and gradient magnitude, respectively. Finally, the invariance to image location, scale, and rotation was checked by pixel neighborhoods and histogram-based statistics. Relevant details are provided in the research of Lowe [34].

$$D(x, y, \sigma) = L(x, y, k\sigma) - L(x, y, \sigma) \tag{1}$$

$$\theta(x, y) = \tan^{-1}\left(\frac{\partial L}{\partial y} \middle/ \frac{\partial L}{\partial x}\right) \tag{2}$$

$$m(x, y) = \sqrt{\left(\frac{\partial L}{\partial x}\right)^2 + \left(\frac{\partial L}{\partial y}\right)^2} \tag{3}$$

After tie points were determined, homogeneous coordinates were used to generate a $3 \times 3$ matrix (H), which describes the spatial translation, scaling, and rotation of feature points in different images. If the feature point set of an image is (x,y,1) and that of another image (x′,y′,1), the mapping relationship between tie points of the two images is H. The relationship can be described using Equation (4), where H is to be calculated. At least four pairs of tie points are required to calculate H [35]. However, the degree of freedom must be zero and a unique solution. In practice, more than four pairs of tie points are often used, resulting in a degree of freedom > 0. Therefore, the match with minimum errors must be identified; this process is known as optimization. Kolor Autopano adopts the random sample consensus (RANSAC) algorithm, as shown in Equation (5), to minimize the errors between H and all tie points [10]. Specifically, random sample consensus randomly selects tie points as inliers to calculate matrix H and evaluate the errors between the matrix and other tie points. Subsequently, these tie points are divided into inliers and outliers before inliers are renewed. Said process is repeated until the matrix H with minimum errors is obtained, serving as the optimal solution. The number of iteration (N) in Equation (5) is chosen to ensure that the probability p (usually set to 0.99) and at least one of the sets of random samples exclude an outlier. Let u indicate the probability that the selected data point is an inlier, and v = 1 − u the probability of observing an outlier. N iterations of the minimum number of points show that m is required.

$$
\begin{bmatrix} x \\ y \\ 1 \end{bmatrix} = \begin{bmatrix} h_{00} & h_{01} & h_{02} \\ h_{10} & h_{11} & h_{12} \\ h_{20} & h_{21} & h_{22} \end{bmatrix} \times \begin{bmatrix} x' \\ y' \\ 1 \end{bmatrix} \tag{4}
$$

$$
1 - p = (1 - u^m)^N \tag{5}
$$

### 3.2.2. Image Warping and Blending

Image warping determines the distortion level of an image in a space by using the tie points identified in homography. One of the selected two images serves as a reference, whereas the other is projected onto the coordinate space according to the reference. After computation and distortion, the projected image is projected onto the reference image, thus achieving image warping. During image projection, the projected image is distorted to enable two images to be superposed. However, during image stitching, excessive distortion of the nonoverlapping area is likely. Cylindrical projection and spherical projection can mitigate such distortion; therefore, we confirmed that said projection methods are suitable for panoramic stitching and projected the images onto a cylinder or sphere.

Let the coordinates of an image be (x,y) and the projected coordinates on a sphere be (x,y,f). The spherical coordinate system is displayed as $(r,\theta,\varphi)$, where r denotes the distance between the sphere center and the target, $\theta$ is the included angle between r and the zenith (range = $[0, \pi]$), and $\varphi$ represents the included angle between the r plane projection line and X-axis (range = $[0, 2\pi]$). The spherical coordinate system can also be converted into a Cartesian coordinate system. Accordingly, conversion between the image and spherical coordinate systems can be described using Equation (6). By converting the spherical coordinate system into a Cartesian one and implementing homography, we achieved image warping.

$$
(r \sin \theta \cos \varphi, r \sin \theta \sin \varphi, r \cos \theta) \propto (x, y, f) \tag{6}
$$

Blending, the last step of image stitching, involves synthesizing warped images by using color-balancing algorithms to create an image gradient on the overlapping area of two images. In this manner, chromatic aberration of the resulting image stitched from multiple images can be minimized. Common methods include feather blending, and multiband blending; please refer to [36] for further details.

### 3.2.3. Accuracy Assessment

Kolor Autopano was then used to calculate the root mean square errors (RMSEs) of the stitched panoramic images, as shown in Equation (7), where N denotes the number of tie points. The resulting RMSE value in this study represents the pixel distance (Diff) between tie points [37]. A value < 5 pixels indicates favorable stitching quality. Conversely, a value ≥ 5 indicates undesirable quality and the possibility of mismatch; in such cases, the tie points should be reviewed.

$$\text{RMSE} = \sqrt{\frac{\sum_{i=1}^{N} Diff_i}{N}} \tag{7}$$

### 3.3. Virtual Tour with VR

Panoramic images of several stations were obtained after image stitching. All the panoramic images in this study were imported into Kolor Panotour and displayed in a virtual tour. In addition to webpage construction, the Kolor Panotour software enables users to add data associated with a point of interest and attribute as well as insert images, videos, and hyperlinks. The software also facilitates the generation of VR navigation environments. Koehl and Brigand [16] provided an introduction to and outlined the application of Kolor Panotour.

### 3.4. Designed Questionnaire

The questionnaire employed in this study comprised four questions, which were rated on a 5-point Likert scale. A higher score indicates satisfaction with or interest in the designed platform. The questionnaire was designed using Google Forms with a quick response (QR) code attached. A total of 62 students completed the questionnaire, and some students were also interviewed. The survey questions are as follows:

Q1.  After using the virtual tour webpages, compared with the scenario where only an introduction is provided by the instructor, can you more easily identify surveying targets on campus?
Q2.  Are you satisfied with the overall webpage design?
Q3.  Are you interested in research on surveying practice courses that employ panorama and VR technology?
Q4.  Do you like courses that incorporate information technologies (e.g., e-learning)?

## 4. Results

### 4.1. Camera Settings

Appropriate exposure and focus are essential for taking a suitable picture [38]. Exposure is determined by the shutter speed, aperture size, and ISO value, whereas the image is in focus and becomes clear only once the lens–object distance is correctly adjusted.

A typical imaging device can only capture an extremely limited range within the complete dynamic range. Consequently, using general imaging devices can lead to a severe loss of scene information, particularly in highlights and shadows [39]. By using bracketing, we took normal, darker, and brighter photos and combined these photos, which have different exposure levels, to obtain a greater exposure dynamic range. After several tests, the suitable parameters for camera settings in the study case are shown in Table 2.

**Table 2.** Used parameters for camera settings in this study.

| Shutter Speed (Sec.) | Aperture Size | ISO Value |
|:---:|:---:|:---:|
| 1/500~1 | F/11~F/16 | 100~400 |

### 4.2. Study Site and Routes

The study site was on the main campus (northwestern side) of Feng Chia University in Taichung City, Taiwan (Figure 4). The two routes (symbols 1 and 2), comprising surveying targets surrounding the Civil/Hydraulic Engineering Building and Science Building, were regarded as elevation-based measurement routes (i.e., the blue and green lines in Figure 4D). The site for angle-based measurement (symbol 3) was located on the lawn, which is represented by red lines in Figure 4D. According to the measurement tasks and targets, a total of 15 stations for panoramic photography were set up (Figure 5).

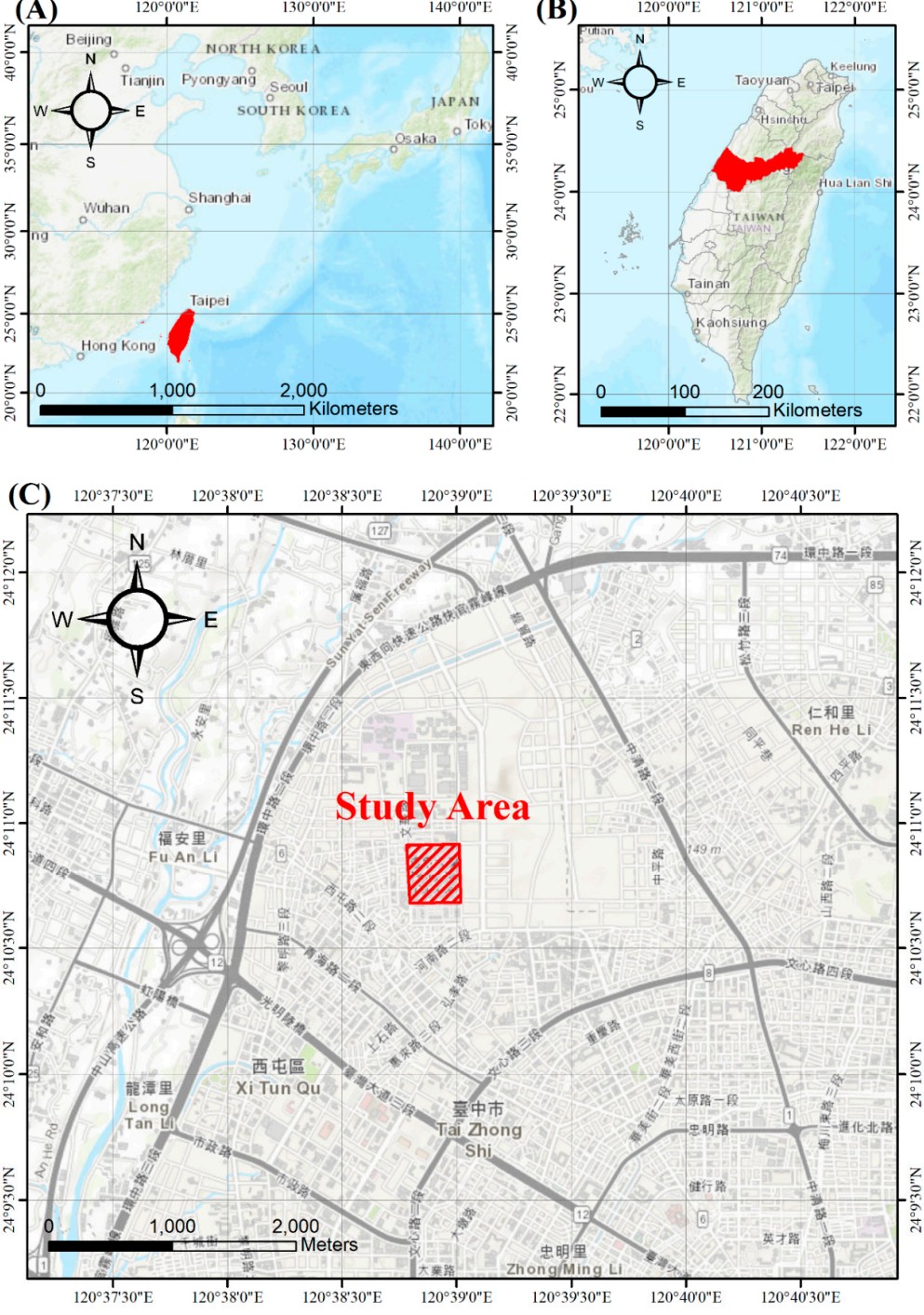

**Figure 4.** *Cont.*

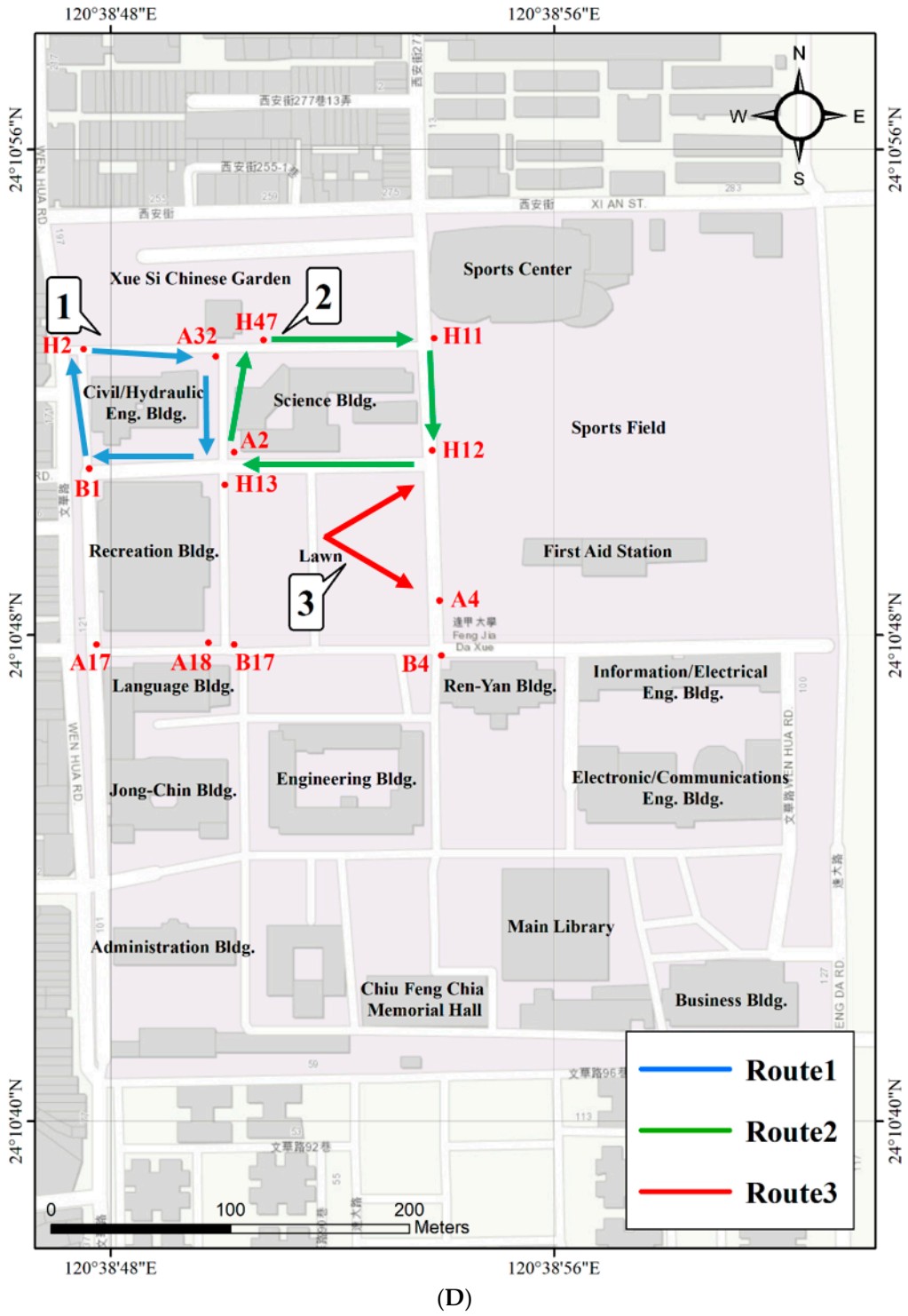

**Figure 4.** Study site (**A–C**), stations and routes (**D**) for the course of Surveying Practice in Feng Chia University (FCU), where blue and green colors indicate the routes for the elevation-based measurement, and red represents the station for the angle-based measurement.

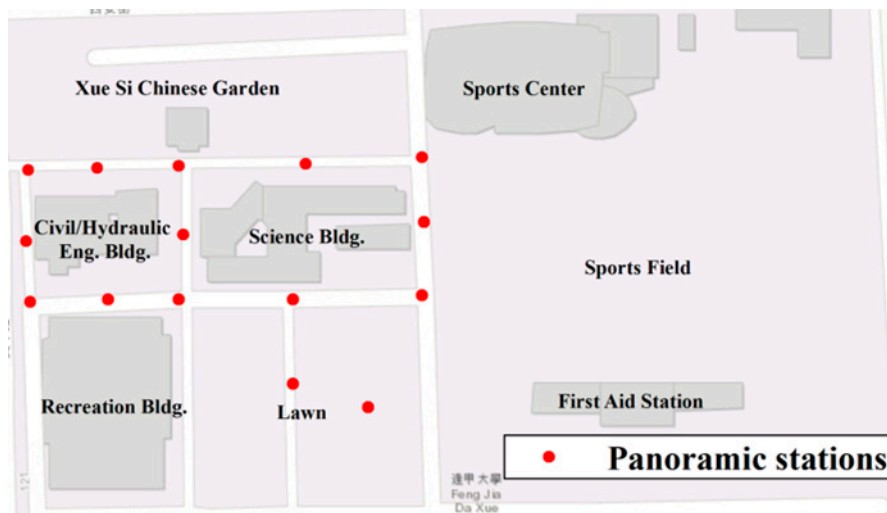

**Figure 5.** Panoramic stations (red points) for capturing images.

*4.3. Developed Platform*

4.3.1. Image Stitching

Figure 6 depicts the stitched image of a surveying station as an example; the green marks denote an RMSE < 5 pixels, whereas the red marks denote stitching errors with an RMSE ≥ 5. This figure demonstrates that most of the tie points meet the suggested standard. However, the stitching quality between trees and the sky was lower because when the camera took pictures, the target object moved, causing image stitching to fail. Therefore, manual adjustment of the tie points or postprocessing of the image was required. Figure 7 exhibits the stitching results of panoramas at four stations. These images were later imported into the virtual tour platform to enable panoramic navigation.

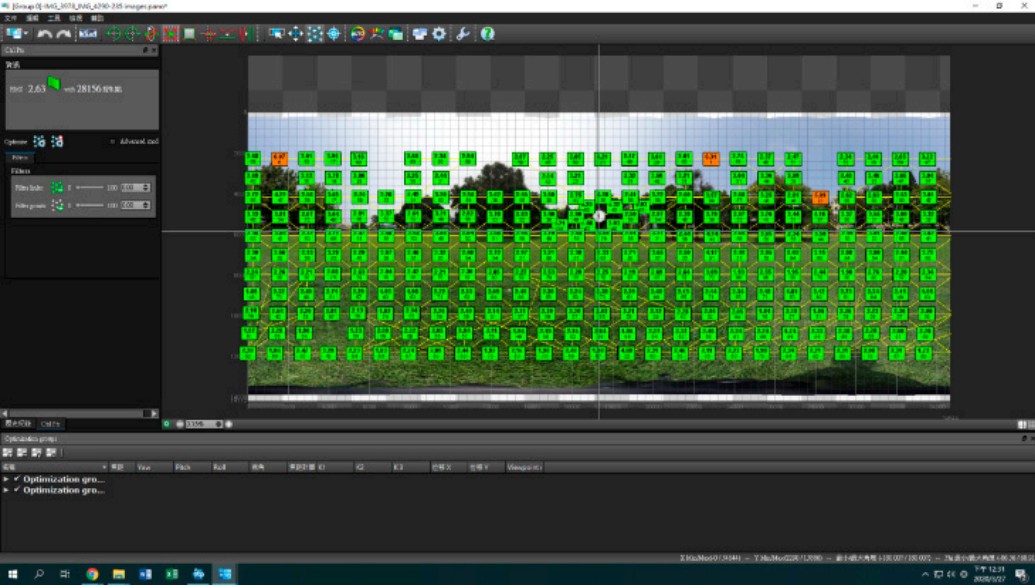

**Figure 6.** An example of image stitching with root mean square errors (RMSEs) where green color represents excellent results.

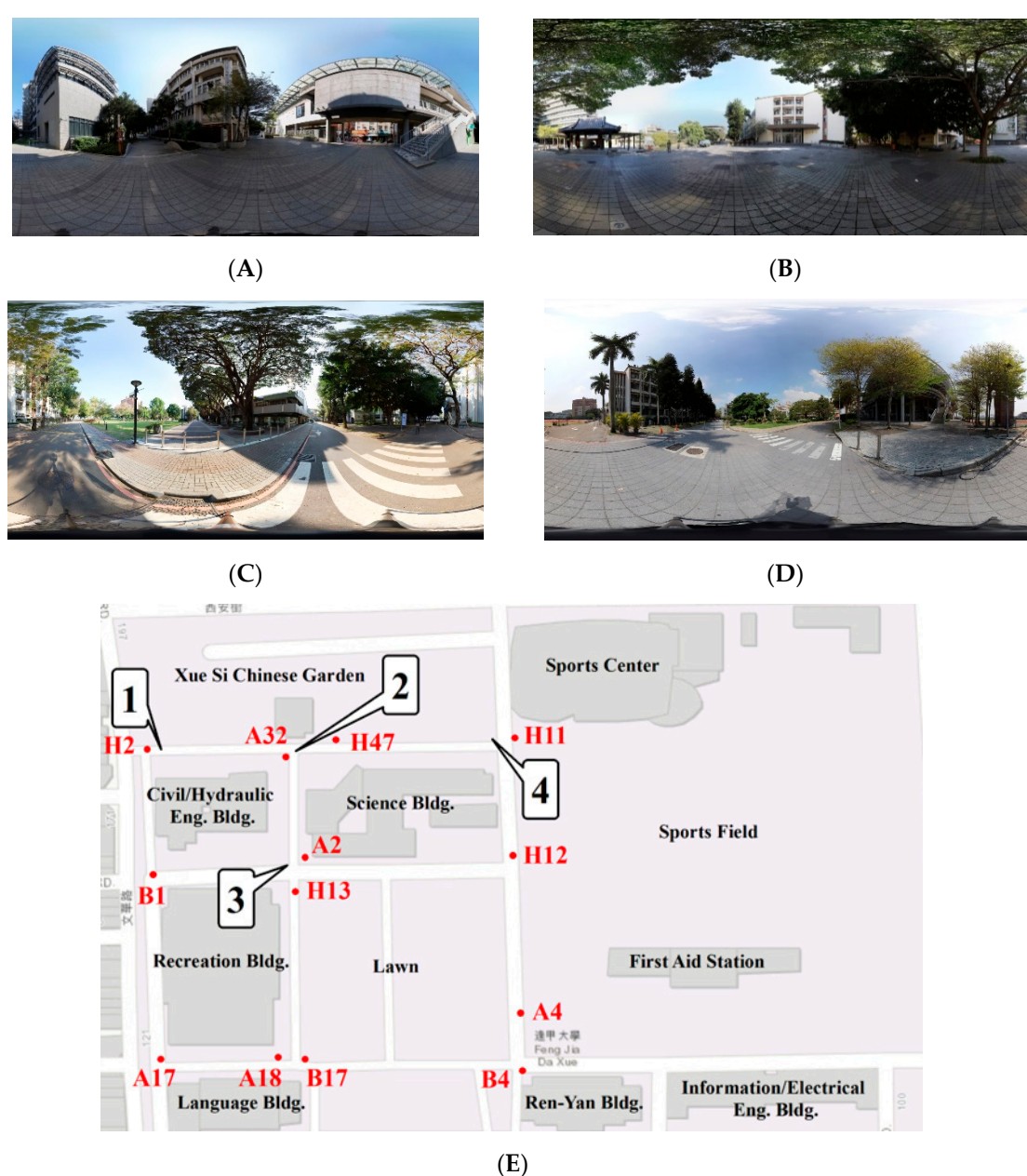

**Figure 7.** Examples of the stitched images, located (**A**) symbol 1, (**B**) symbol 2, (**C**) symbol 3, and (**D**) symbol 4 in (**E**).

4.3.2. Webpage and Virtual Tour

Figure 8 illustrates the designed webpage framework and panoramic stations, which have a spatial relationship that is consistent with that in Figure 5 from the top view. The homepage displays the panorama of Station 14. When users visit Station 15, they can read instructions on angle-based measurement and watch a tutorial on angular measuring device setup. Station 11-3 features instructions on elevation-based measurement and a tutorial on relevant instrument setup. This station also provides on-site views of the two routes available for surveying. Next, the research team connected the elements in Figure 8 to a virtual tour platform to establish a webpage, where various information can be added, including points of interest and attributes. Designers could also insert pictures, videos, and hyperlinks (Figure 9).

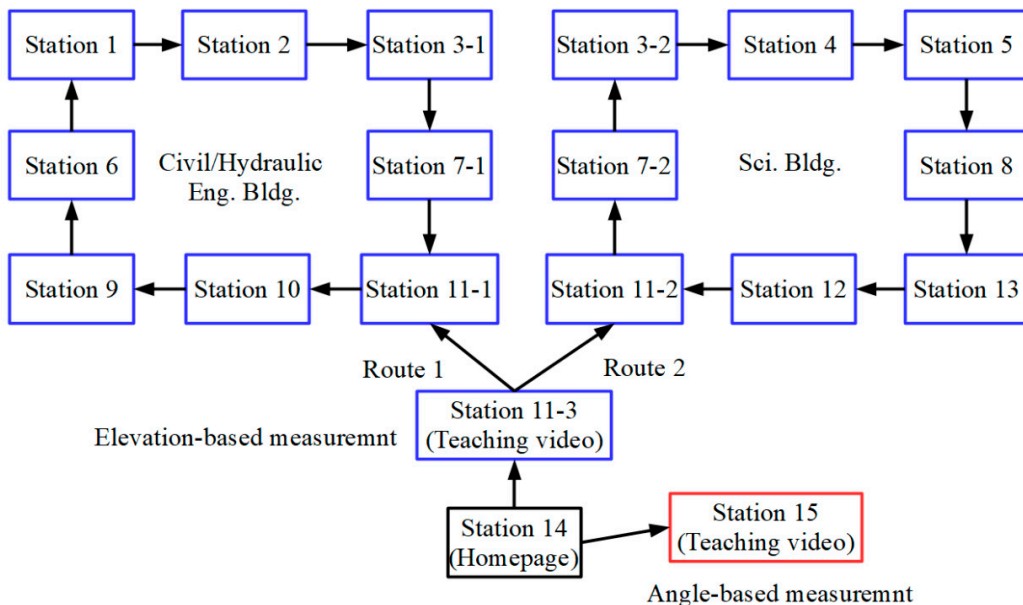

**Figure 8.** Webpage framework and panoramic stations, where blue and red colors represent elevation- and angle-based measurements.

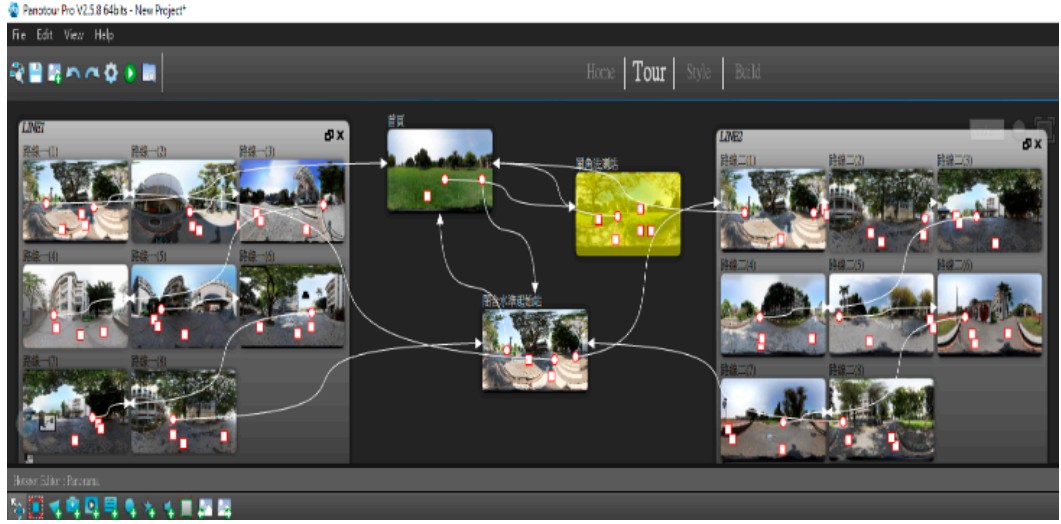

**Figure 9.** An example of setting a virtual tour.

### 4.3.3. Demonstration

The platform homepage is displayed in Figure 10. On the upper left (symbol 1) is a link to Feng Chia University's website; control bars (symbol 4) are located on the bottom left, enabling the user to zoom in, zoom out, or switch the image to VR mode. At the bottom middle (symbol 5) are relevant instructional documents, and on the bottom right (symbol 6) is a quick map showing the navigation direction. The homepage hyperlink is located in the upper right corner (symbol 3). By navigating around the homepage 360° panorama, the user can locate the entrances for tutorials on elevation-based and angle-based measurement (Figures 11A and 12A).

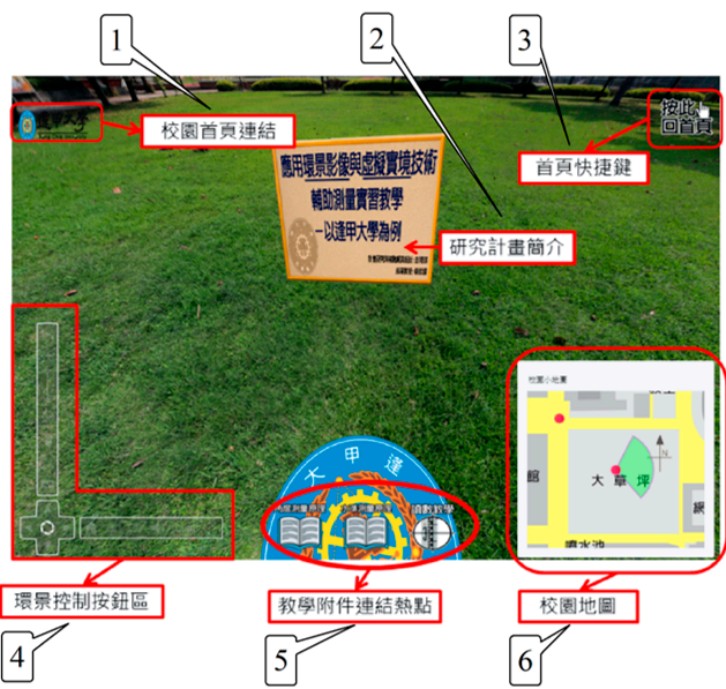

**Figure 10.** Homepage of the developed platform. Symbol 1: hyperlink to the FCU official webpage; symbol 2: information of the supporting project; symbol 3: back to the homepage control bar; symbol 4: control bars (zoom in, zoom out, or switch the image to VR mode); symbol 5: hyperlink to the related documents; symbol 6: quick map.

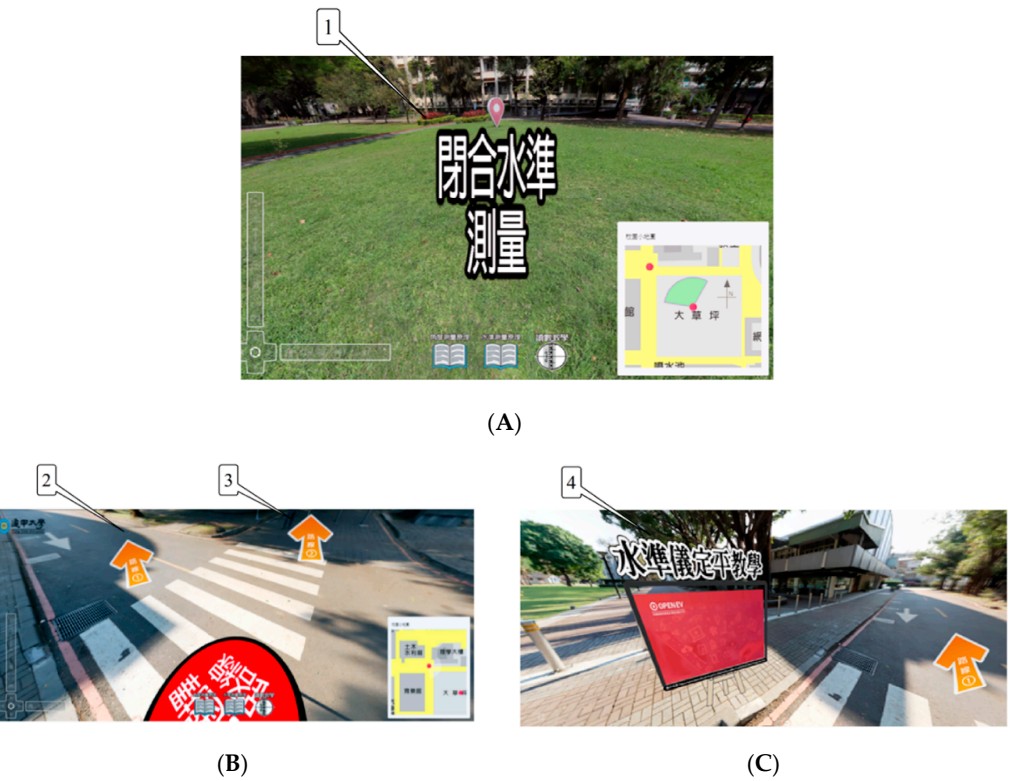

**Figure 11.** Webpage for teaching the elevation-based measurement. (**A**) Entrance for angle-based measurement (symbol 1, or called leveling measurement); (**B**) Route selection (symbols 2 and 3 represent routes 1 and 2, respectively); (**C**) Teaching video (symbol 4) for setting up the instrument.

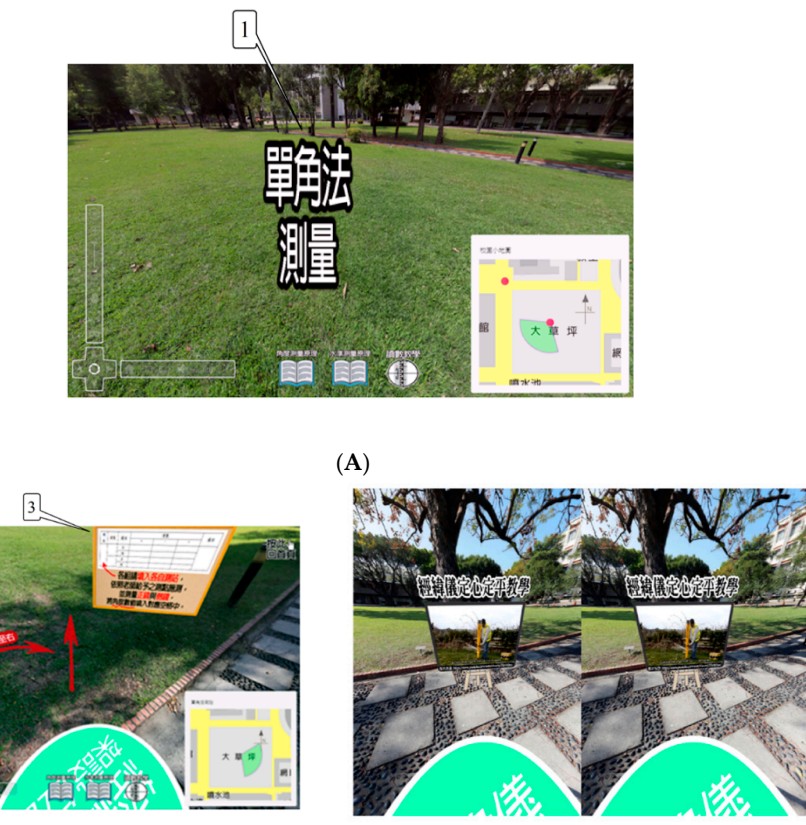

**Figure 12.** Webpage for teaching the angle-based measurement. (**A**) Entrance for angle-based measurement (symbol 1); (**B**) Aimed direction from left for measurement (symbol 2) and how to record the observations (symbol 3); (**C**) Teaching video for setting up the instrument with VR mode.

After clicking on the entrance for elevation-based (or called leveling) measurement, users have two routes (symbols 2 and 3) to choose from in Figure 11B. Additionally, a tutorial video (symbol 4) on instrument setup for elevation-based measurement is available at Station 11-3 (Figure 11C). Figure 12b depicts the webpage shown when users click on the entrance for angle-based measurement instructions. This page not only instructs users on how to record surveying measurements (symbol 3), aim targets, and set the surveying direction (symbol 2), but also teaches them how to set up the measuring instrument in a tutorial video (Figure 12C). By clicking the VR icon on the control bars, the image switches to VR mode. Users can then place their smartphone inside a portable VR viewer, connect a joystick to the system, and enjoy the VR tour (Figure 12C).

*4.4. Questionnaire-Based Results*

We received 62 questionnaire responses from the participating students; the statistical results are listed in Table 3. A higher score indicates stronger approval of the use of the designed platform and information technologies in teaching. Q1 and Q2 aimed to investigate the effectiveness of the designed platform in teaching as well as its content display, and Q3 and Q4 attempted to determine whether the use of information technologies helps strengthen learning interest. Most of the ratings for these four questions were > 4 (92% of the total responses), signifying positive feedback from respondents. All the students used the web-based platform; only some of the students tested the head-mounted equipment because of limited devices. The major problem using a head-mounted instrument is the layout for display. The developed platform was revised based on feedback, such as adjusting the size of the quick map and adding other targets for measurement.

**Table 3.** Results of the questionnaire.

| (%) | Score | | | | |
|---|---|---|---|---|---|
| | **5** | **4** | **3** | **2** | **1** |
| Q1 | 53 | 44 | 3 | - | - |
| Q2 | 48 | 44 | 8 | - | - |
| Q3 | 37 | 60 | 3 | - | - |
| Q4 | 45 | 55 | - | - | - |

## 5. Discussion

### 5.1. Capturing Modes

To capture a high-quality image, one should observe the surrounding light, estimate the area of the surrounding environment, and adjust camera parameters (e.g., shutter, aperture, ISO value, and bracketing level) accordingly. By increasing the rate of overlap in the panoramic instrument, we expanded the overlapping area of adjacent photos, increased the number of tie points, and reduced the RMSEs, thereby enhancing the success rate of image matching.

In case of undesirable matching results based on GigaPan capturing, the problem must be identified in the photos. If overexposure or underexposure is confirmed, the aforementioned bracketing and high-dynamic range mode can be selected to overcome difficulties in feature point extraction caused by exposure-related problems. If the scene in question has few feature points, we recommend taking a new photo to increase the rate of overlap. Therefore, a larger overlapping area can be obtained with corresponding increases in the numbers of feature and tie points.

For example, Figure 13 exhibits a stitched panorama. Table 4 lists the capturing time required and post stitching RMSEs at different rates of overlap. When the targets are far from the viewer's perspective in a scene, the rate of overlap contributes less to the stitching quality. However, when some of the targets are far from the viewer's perspective and others are nearer, the rate of overlap can positively influence the stitching quality. Furthermore, the rate of overlap is directly proportional to capturing time.

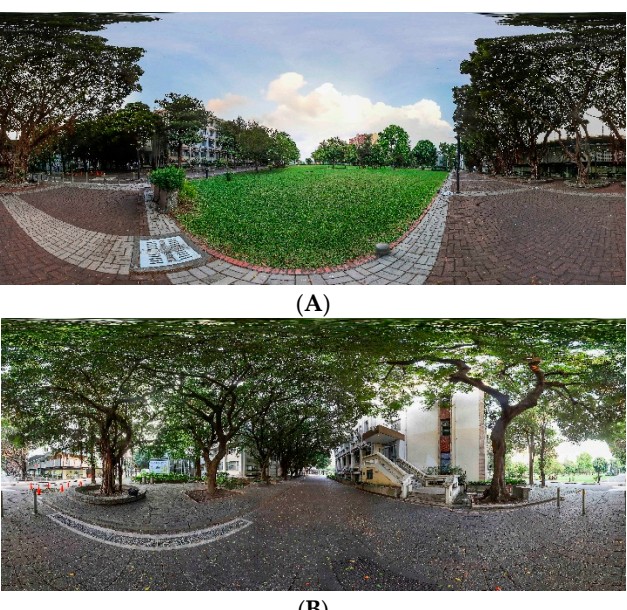

(**A**)

(**B**)

**Figure 13.** Cases for comparing different overlap ratios and RMSEs after image stitching. (**A**) Case 1; (**B**) Case 2.

**Table 4.** Statistics of Figure 13 for comparing different overlap ratios and root mean square errors (RMSEs).

| Overlap | Capturing Time (Min) | RMSE (Pixel) | |
| --- | --- | --- | --- |
| | | Case 1 | Case 2 |
| 30% | ~3 | 3.39 | 3.48 |
| 50% | ~5 | 3.37 | 3.39 |
| 70% | ~9 | 3.35 | 3.29 |

In outdoor photography, objects often move, causing ghosting effects or mismatches during image stitching (e.g., characters or clouds moving and grass moving in the wind) in the case of GigaPan capturing. To address this problem, we recommend on-site investigation and planning in advance; for example, pictures could be taken when atmospheric pressure differences are small or when fewer people are in the area. Alternatively, the effects of object movement can be mitigated by adjusting the shooting sequence of the panoramic instrument. For instance, letting the camera shoot vertical photos before moving to the left or right (Column-left/right) can reduce the ghosting effects of characters. The shooting process can also be paused and continue after people pass by the target site. Row-down shooting after the camera takes a 360° horizontal shot also helps avoid problems related to clouds and sunlight; the taken photos can later be adjusted using postproduction software.

### 5.2. Contributions, Comparison and Limitations

There are two approaches for capturing a panoramic image in general. One is to use a 360-degree spherical camera. Another is to adopt a platform with a camera for capturing and stitching the images, such as GigaPan. The former can easily and simply create a panoramic image, but the resolution is not better than the latter, and there is no chance to improve the image quality. On the other hand, the platform-based method requires stable conditions to capture images. This is a trade-off between these operations for producing a panoramic image. High resolution [40] is necessary in this study to display the targets for measurement in the field. Thus, the GigaPan-based approach was chosen in this study case.

This study contributes to knowledge by (a) discussing panoramic photography and image stitching quality as well as providing suggestions on camera parameters and GigaPan settings, (b) proposing strategies for the production of panoramic images that are more operable and have a higher resolution than Google Street View, (c) using information technologies (i.e., virtual tour tools and VR) to develop an assistance platform for teaching, (d) applying the designed platform to an engineering course, and (e) assessing teaching effectiveness through a questionnaire survey.

Ekpar [41] proposed a framework for the creation, management, and deployment of interactive virtual tours with panoramic images. Based on this concept, many cases for displaying campuses [42,43], cathedrals [44], and culture heritage sites [45] were explored in the previous literature. This study not only visualized reality-based scenes using panorama- and virtual tour-based technologies, but also connected related documents for engineering education. Furthermore, the teaching site was also emphasized in this study. E-learning is a trend of education; it can help teachers reduce the load of teaching and further concentrate on the professional issues in subjects. Furthermore, this study provided a useful platform to help students who cannot come to the classroom because of special circumstances (e.g., COVID-19).

In terms of limitations, before students could use the platform designed in this study, instructors had to explain relevant theories and provide detailed instructions for equipment operations. If students lack basic understanding of surveying, their learning outcomes might not meet expectations. Additionally, we did not recruit a control group and thus, could not compare the learning results between students who used the designed platform and those who did not. We endeavor to test and verify the effect of the designed platform on learning outcomes in future research. The questionnaires and assessments

for improving engineering education could be designed to put more emphasis on user testing and the responses from students, for example,

1. Sampling design for statistical tests:

   - Testing differences of final grades between student groups with the IT (Information Technologies)-based method and without the IT-based method.
   - Grouping samples by the background of sampled students.

2. Asking more aspects for comprehensive assessment:

   - "Would you recommend this to your friends and colleagues?" followed by "What points do you recommend/not recommend?"
   - "How long did you take to complete the IT-based program?"
   - "Does the virtual tour seamlessly/comfortably guide you?"
   - "Does the virtual tour sufficiently represent the real world?"

3. Comparing and exploring the problems on IT-based and traditional learning.

## 6. Conclusions

To create a multimedia platform that assists students in a surveying practice course, we initially took multiple overlapping images and stitched them into panoramas; subsequently, we used information technologies including virtual tour tools and VR. A full-frame digital single-lens reflex camera with an ultra-wide-angle zoom lens was mounted on a GigaPan panoramic instrument to obtain a 360° horizontal field of vision. The effectiveness of said visualization and information technology application was verified through a questionnaire survey.

The research results indicated that the RMSEs of stitched images were mostly < 5 pixels, signifying favorable stitching quality. The designed platform also features elevation-based and angle-based measurement instructions as well as instrument setup tutorials and documents as supplementary materials. A total of 15 panorama stations were set up for students to navigate. Of the 62 survey respondents, more than 92% agreed that the information technology-based platform improved their engagement in the surveying practice course. Moreover, we discussed and explored the improvement of panoramic image quality (RMSEs), camera parameter settings, capturing modes, and panoramic image processing, as shown in Section 4.1 and Section 5.1. In the future, we plan to compare the learning outcomes in students who used the designed platform (experimental group) with those who did not (control group).

**Author Contributions:** Jhe-Syuan Lai conceived and designed this study; Jhe-Syuan Lai, Yu-Chi Peng, Min-Jhen Chang, and Jun-Yi Huang performed the experiments and analyzed the results; Jhe-Syuan Lai, Yu-Chi Peng, and Min-Jhen Chang wrote the paper. All authors have read and agreed to the published version of the manuscript.

**Funding:** This study was supported, in part, by the Ministry of Education in Taiwan under the Project B from June 2019 to February 2021 for New Engineering Education Method Experiment and Construction.

**Acknowledgments:** The authors would like to thank Fuan Tsai and Yu-Ching Liu in the Center for Space and Remote Sensing Research, National Central University, Taiwan, for supporting the software and panoramic instrument operation.

**Conflicts of Interest:** The authors declare that they have no conflict of interest.

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
