# Peer review of "Panoramic Mapping with Information Technologies for Supporting Engineering Education: A Preliminary Exploration"

_ijgi, doi:10.3390/ijgi9110689_

Round 1

Reviewer 1 Report

The authors presented a method using VR for educating students in civil engineering, which could be a breakthrough in hardness for entry levels to understand survey processes in more intuitive ways. Although the topic is fitting to the fields of this journal, I think the manuscript lacks sufficient analysis ineffectiveness of the VR-based education method. Therefore, I would recommend "Reject" to this submission. I would comment on this manuscript for future submissions.

1. It requires sampling design for statistical tests like below.
- Testing differences of final grades between student groups with the VR-based method and without the VR-based method.
- Grouping samples by the background of sampled students; for example, the n-th year in the school, courses are taken in the past.

2. Questionaire should ask more aspects for comprehensive assessment such as:
- "Would you recommend this to your friends and colleagues?" followed by "What points do you recommend / not recommend?"
- "How long did you take to complete the VR-based program?"
- "Does the virtual tour seamlessly/comfortably guide you?"
- "Is the virtual tour sufficiently represent the real world?"
The questions need to be specified through discussions of the problems on VR-based learning and visual recognition.

Thank you in advance for your consideration in incorporating my comments to developing the paper.

Reviewer 2 Report

The abstract concerns an integrated multimedia platform aimed to assist students in a surveying practice course.

Unfortunately it is not clear the platform description neither it's design modifications according to respondents’ feedback.

Instead, a virtual tour along the campus has been developed, giving a virtual support to present a video about the virtual tour technique, detached by what the user is viewing: a campus view. "The expectation of a tour is that it will provide richer content and detail about the subject matter" (Campus virtual tour, aa.vv., IJACET vol.6, Issue 4, April 2017)

The article focuses on stitching, but it is nowadays a simple software task; it would be better to give troubleshooting tips during the process: for example, what about sunny and cloudy area in the images to stitch as well as bright and dark areas, the presence of moving objects/people...

For this application context-detached, it would be more simpler, cheaper and faster to use 360 Degree Spherical Camera with Dual Sensors, which have firmware stitching. 

Reviewer 3 Report

Line 81 - check the spelling.

Figure 1 - The conceptual procedure graph should highlight the feedback offered by questionnaires to improve the platform design.

Section 3.2 - it is not clear if you developed an "in-house" solution for Image stitching" or you adopted a commercial software (Kolor Autopano?). If the latter, it is ok to recap how are the general principles used for the process, but you should clarify. 

Line 206 - Too general reference (proceedings of the conference)

Section 4.1 - when working outdoor - as in this study - the camera setting optimization mainly depends on light/weather conditions; it is impossible to "recommend" general settings. Indeed values in Table 2 describe a great variability of settings.

Section 4.3.2. Webpage and Virtual Tour - I cannot find the Stations mentioned in either figure 5 or figure 7.

Caption Figure 10: delete the last word

Questionnaire and Discussion - All the users experienced the platform both via web and with a head-mounted display? Did you record any cons in the use of the head-mounted display (dizziness, headache, nausea)?   

Reviewer 4 Report

This article presents a virtual tour platform to create webpages and a virtual reality environment. For that, this platform takes multistation-based panoramic images and imported the processed images. The platform aims to assist students in a surveying practice course.

I would like to add that the proposed idea is interesting and could be a starting point for future virtual tour platforms. However, It's difficult for me to appreciate the contribution. I recommend that the authors carry out a restructuring of the article to improve the contribution presentation.

Some items that should be clarified and revised:

-The methodology section provides the article contribution. However, the authors present a mathematical formulation limited. This formulation is essential since it allows us to replicate the results presented and the implementation.

-The discussion section allows us to visualize the scope of the results and its comparison with previous work. However, the authors omit the comparison with previous work. I recommend that the authors add this discussion.

-Also, since the platform is novel, I recommend that in the discussion section, the authors should argue and elucidate how the work would concretely impact the community of the virtual tour platforms.

Round 2

Reviewer 1 Report

I carefully reviewed and confirmed my comments have been addressed in the revision.

Reviewer 2 Report

Dear Authors, thank you for the work done to improve your article.

I wrote that the video about the virtual tour technique is detached by what the user is viewing, underlining that "The expectation of a tour is that it will provide richer content and detail about the subject matter": this issue has been fixed only adding its reference in your work.

The main issue is again unresolved: the virtual tour goal is to reproduce the real world improving its knowledge through multimedia information linked to hotspots selected according to the work purpose.

In your virtual tour the multimedia information available is "a tutorial video (symbol 4) on instrument setup for elevation-based measurement" located in the low left corner of the webpage shown in figure 4.

This video is not giving information about the objects in the virtual tour scene. The video could be seen in any other scene, context or tool - It is for me conceptually wrong, sorry!

Reviewer 4 Report

This article presents a virtual tour platform to create webpages and a virtual reality environment. For that, this platform takes multistation-based panoramic images and imported the processed images. The platform aims to assist students in a surveying practice course.

I would like to add that the proposed idea is interesting and could be a starting point for future virtual tour platforms. On the other hand, the authors restructured the article with my suggestions and recommendations.